# DNA Polymerase θ: A Cancer Drug Target with Reverse Transcriptase Activity

**DOI:** 10.3390/genes12081146

**Published:** 2021-07-27

**Authors:** Xiaojiang S. Chen, Richard T. Pomerantz

**Affiliations:** 1Molecular and Computational Biology, USC Dornsife Department of Biological Sciences, University of Southern California, Los Angeles, CA 90089, USA; xiaojiac@usc.edu; 2Department of Biochemistry and Molecular Biology, Sidney Kimmel Cancer Center, Thomas Jefferson University, Philadelphia, PA 19107, USA

**Keywords:** DNA polymerase, reverse transcriptase, RNA, reverse transcription, double-strand break repair, translesion synthesis

## Abstract

The emergence of precision medicine from the development of Poly (ADP-ribose) polymerase (PARP) inhibitors that preferentially kill cells defective in homologous recombination has sparked wide interest in identifying and characterizing additional DNA repair enzymes that are synthetic lethal with HR factors. DNA polymerase theta (Polθ) is a validated anti-cancer drug target that is synthetic lethal with HR factors and other DNA repair proteins and confers cellular resistance to various genotoxic cancer therapies. Since its initial characterization as a helicase-polymerase fusion protein in 2003, many exciting and unexpected activities of Polθ in microhomology-mediated end-joining (MMEJ) and translesion synthesis (TLS) have been discovered. Here, we provide a short review of Polθ‘s DNA repair activities and its potential as a drug target and highlight a recent report that reveals Polθ as a naturally occurring reverse transcriptase (RT) in mammalian cells.

## 1. Introduction

Mutations in homologous recombination (HR) genes *BRCA1* and *BRCA2* are strongly predisposed to breast and ovarian cancer [1,2,3,4,5,6]. Since BRCA deficient cancer cells are impaired in HR, they are highly susceptible to DNA damage compared to normal cells [4,5]. Drugs that cause DNA damage or inhibit DNA repair, such as Poly (ADP-ribose) polymerase 1 (PARP1) inhibitors, can therefore cause synthetic lethality in BRCA deficient cells while sparing normal cells [4,7,8,9]. Highly anticipated PARP inhibitors (PARPi), however, lead to drug resistance, which often causes patient mortality [7,10,11,12]. Thus, it remains important to identify and develop alternative drug targets involved in DNA repair for BRCA deficient cancers that reduce drug resistance and potential side effects.

Studies performed in 2015 identified the multi-functional DNA repair protein DNA polymerase θ (Polθ) as a promising drug target in HR-deficient cancers [13,14]. Polθ is upregulated in the majority (70%) of breast tumors and epithelial ovarian cancers [14,15,16,17,18], and its overexpression correlates with HR defects and a poor clinical outcome [14,15,16,19]. Polθ also confers resistance to ionizing radiation, genotoxic chemotherapy drugs (e.g., topoisomerase inhibitors, cisplatin), and PARPi [14,20,21,22,23]. Thus, in addition to promoting the proliferation of HR deficient cells, Polθ’s DNA repair activities, such as microhomology-mediated end-joining (MMEJ) of double-strand breaks (DSBs), contribute to cellular resistance of a variety of genotoxic anti-cancer agents.

## 2. Overview of Polθ DNA Repair Activities

Polθ is a large multi-functional protein containing an N-terminal superfamily 2 (SF2) helicase (Polθ-hel) [24], an unstructured central domain, and a C-terminal A-family polymerase domain (Polθ-pol) that is structurally similar to bacterial Pol I enzymes, such as Klenow fragment and Thermus aquaticus (Taq) Pol (Figure 1A) [25,26,27]. However, in contrast to related Pol I enzymes, Polθ-pol is highly error-prone and is promiscuous in regards to its use of nucleic acid and nucleotide substrates [17,28,29,30,31]. For example, despite being an A-family polymerase that typically possess relatively high fidelity DNA synthesis, Polθ-pol possesses a deficient proofreading domain due to acquired mutations, and carries out translesion synthesis (TLS) opposite various DNA lesions in vitro and in cells, and therefore is involved in DNA damage tolerance (Figure 1B) [17,31,32]. Recent studies demonstrate that Polθ confers resistance to ultraviolet (UV) light-induced intrastrand DNA crosslinks via its error-prone TLS activity [33]. The only other A-family mammalian Pol known to exhibit error-prone DNA synthesis and accommodate template lesions is Polν, which also lacks proofreading activity [34]. The majority of TLS Pols belong to the Y-family of Pols such as Polκ, Polη, and Polι, which possess more solvent-exposed active sites and are also deficient in proofreading [35].

Polθ additionally facilitates DSB repair via MMEJ—also referred to as alternative end-joining (alt-EJ) and polymerase theta mediated end-joining (TMEJ) (Figure 1C) [13,21,25,36,37,38]. For example, Polθ-pol specifically facilitates MMEJ of DNA with 3′ overhangs containing short tracts (2–6 bp) of microhomology in vitro and is essential for MMEJ in cells [13,21,25,36,38]. Recent studies additionally demonstrate that Polθ dependent MMEJ promotes the repair of 5′ DNA-protein crosslinks, which can form as a result of etoposide-induced covalent trapping of topoisomerase 2 onto DNA [20]. These studies explain how Polθ and collaborating MMEJ factors, such as Mre11-Rad50-Nbs1 (MRN), confer resistance to etoposide and potentially other DNA-protein crosslinking agents, such as topoisomerase 1 inhibitors. Another recent report found that Polθ-pol unexpectedly exhibits DNA endonuclease activity, which is implicated in end-trimming during MMEJ [39]. Prior studies also discovered 5′-deoxyribose phosphate lyase activity exhibited by the polymerase domain, suggesting a possible function in base excision repair [40].

Polθ has also been implicated in the repair of replication-associated DNA breaks, which is supported by its semi-synthetic lethal interaction with ATR [41]. Prior studies demonstrated a semi-synthetic lethal interaction between Polθ and ATM, which provided early evidence in support of Polθ activity in DSB repair [42]. Lastly, recent studies reveal a new function for Polθ in suppressing mitotic crossovers following DSBs, which preserves genome integrity by suppressing the loss of heterozygosity [43,44]. Carvajal-Garcia et al. additionally found that Polθ is synthetic lethal with Holliday junction resolvases (SLX4, GEN1), which supports a model whereby Polθ acts on HR intermediates [43]. Although more mechanistic studies are needed to fully elucidate how Polθ suppresses mitotic crossovers, Carvajal-Garcia, J. et al. suggest that Polθ acts upstream from Holliday junction resolvases on HR intermediates where it can potentially perform MMEJ [43].

## 3. Polθ Enzymatic Domains as Drug Targets

Although MMEJ is active during S-phase, DSB repair is primarily performed by the more accurate HR DSB repair pathway that relies on BRCA1, BRCA2, and associated proteins (i.e., PALB2, RAD51, RAD51 paralogs) (Figure 2A) [45,46,47,48]. The DNA nuclease complex MRN along with CTBP-interacting protein CtIP is involved in the initial DSB 5′-3′ resection process that is required for the generation of 3′ ssDNA overhangs that support HR and MMEJ (Figure 2A) [48,49]. This suggests competition exists between MMEJ and HR pathways for acting on 3′ ssDNA overhangs [48]. Since MMEJ can serve as a backup repair pathway in HR-deficient cells (Figure 2B), suppression of Polθ in BRCA deficient cells causes synthetic lethality but has little or no effect in BRCA proficient cells (Figure 2C) [13,50]. For example, suppression of *POLQ* gene expression in *BRCA1* mutant (BRCA1-mut) and *BRCA2* mutant (BRCA2-mut) cells resulted in a severe reduction in cell survival [13,14,51] (https://depmap.org/portal/ccle/ accessed on 1 May 2021), whereas suppression of *POLQ* in BRCA wild-type (BRCA-WT) cells had little or no effect [13,14,51]. Consistent with this, BRCA-deficient cancer cells were shown to be more dependent on Polθ expression for their survival in the presence of genotoxic agents and PARPi [14,23]. This synthetic lethal relationship between Polθ and HR was also demonstrated in mouse models [14].

Intriguingly, the DNA synthesis and ATPase activities of Polθ-pol and Polθ-hel, respectively, were shown to promote the survival of BRCA1-mut cells [14,52]. For example, respective inactivation of these domains via site-specific CRISPR/Cas9 mutagenesis within the endogenous *Polq* gene significantly reduced colony formation of BRCA1-deficient cells versus BRCA1-proficient cells [52]. This indicates that pharmacological inhibition of either Polθ enzymatic domain will kill BRCA1 deficient cancer cells while having little or no effect on normal cells. CRISPR/Cas9 knock-out studies also showed that Polθ is synthetic lethal with BRCA2 and other DNA repair factors (i.e., RAD54, Ku70/80, FANCJ) [38,51,53]. Lastly, *POLQ* gene suppression appears to be additive or synergistic with PARPi and DNA damaging agents in killing HR defective cells [14]. Consistent with this, a recently reported potent and selective Polθ polymerase (Polθ-pol) inhibitor developed by Artios Pharma acts additively with PARPi in HR-deficient cells and selectively kills HR-deficient cells as a single agent at low micromolar concentrations [54]. Because Polθ confers resistance to IR and topoisomerase inhibitors (i.e., etoposide, camptothecin) [20,22], Polθ inhibitors are also expected to reduce cellular resistance to these widely used cancer therapies that also cause DNA breaks and replicative stress. Indeed, Artios Pharma’s Polθ-pol inhibitor significantly reduced cellular resistance to IR [54]. Polθ inhibitors are also likely to act synergistically with platinum-based chemotherapy agents in HR-deficient cells based on genetic studies [23].

A more in-depth understanding of the respective enzymatic functions of Polθ-pol and Polθ helicase (Polθ-hel) in MMEJ and potentially other DNA repair mechanisms will better inform strategies for successfully developing this multi-functional protein as a therapeutic target. For example, although Polθ-pol likely has a single function in MMEJ (extension of minimally paired ssDNA overhangs) (Figure 1C), the helicase appears to promote MMEJ dependent and independent DNA repair activities. Ceccaldi et al. reported that Polθ-hel interacts with RAD51 and suppresses toxic RAD51 intermediates to confer a survival advantage in BRCA-deficient cells (Figure 3A) [14]. However, it was subsequently found that genetic mutation of the previously characterized Polθ-hel RAD51 binding site had no effect on the survival of BRCA1-deficient cells [52]. Thus, further research is needed to confirm the functional relevance of reported Polθ-hel:RAD51 interactions. We recently demonstrated that Polθ-hel exhibits ATP-dependent DNA unwinding activity with 3′–5′ polarity, similar to RECQ type SF2 helicases (Figure 3B) [55]. However, whether this activity contributes to MMEJ or other DNA repair activities remains unclear. We previously found that Polθ-hel utilizes the energy of ATP hydrolysis to dissociate RPA from ssDNA overhangs that enables ssDNA annealing and stimulation of MMEJ (Figure 3C) [52]. The ability of RPA:ssDNA complexes to suppress MMEJ in mammalian cells is consistent with a previously reported role for RPA in suppressing a different form of microhomology-mediated DSB repair in yeast [56].

More recently, we found that Polθ-hel strongly contributes to MMEJ in a purified system by cooperating with Polθ-pol on ssDNA overhangs, which was independent of its ATPase function [25]. For example, although Polθ-pol can perform MMEJ of DNA substrates with short 3′ ssDNA overhangs, it is deficient in MMEJ of substrates with long 3′ ssDNA overhangs as a result of its prominent snap-back replication activity on ssDNA (Figure 4A) [25]. So-called snap-back replication results from the enzyme’s unique ability to search for microhomology upstream from the 3′ terminus of ssDNA in cis [25,36]. This enables Polθ-pol to form a stem-loop structure resembling a minimally base-paired primer-template that is efficiently extended by the enzyme (Figure 4A) [25]. Full-length Polθ, on the other hand, exhibited minimal snap-back replication activity on long 3′ ssDNA overhangs and instead primarily performed MMEJ of long ssDNA substrates with 3′ terminal microhomology (Figure 4B) [25]. Moreover, an ATPase mutant version of full-length Polθ, and a Polθ-hel-Polθ-pol fusion protein, which includes a short flexible linker instead of the long central domain, also performed efficient MMEJ of substrates with long 3′ ssDNA overhangs [25]. Combining Polθ-pol and Polθ-hel as separate domains in trans, however, failed to promote MMEJ of long 3′ ssDNA overhangs and instead primarily resulted in Polθ-pol snap-back replication [25]. These biochemical structure function studies indicate that close cooperation between physically linked Polθ-pol and Polθ-hel is required for MMEJ of DNA with long 3′ ssDNA overhangs, and that Polθ-hel ATPase activity is not essential for MMEJ, at least in this minimal purified system that lacks RPA and other auxiliary DNA repair factors. Overall, these studies indicate the Polθ-hel domain binds ssDNA upstream from Polθ-pol, which effectively blocks the polymerase’s prominent snap-back replication activity, and as a result, enables Polθ-pol intermolecular microhomology search as a major mode of action (Figure 4B).

Taken together, multiple lines of evidence demonstrate that both Polθ enzymatic domains contribute to MMEJ and the survival of HR-deficient cells. Consistent with this, small-molecule inhibition of either domain causes the preferential killing of BRCA-deficient cells [54,57]. For example, in addition to the synthetic lethal effects observed for Artios Pharma’s Polθ-pol inhibitor, a recent paper indicates that small-molecule inhibition of Polθ-hel also induces synthetic lethality in BRCA-deficient cells [57]. On the other hand, simultaneous inactivation of both domains may have the largest therapeutic index in BRCA-deficient cells and potentially other DNA repair defective cells, such as those deficient in non-homologous end-joining (NHEJ). In support of this, all of the reported synthetic lethal studies on Polθ used either CRISPR/Cas9 or shRNA to respectively knock-out or strongly suppress the expression of Polθ, which is equivalent to simultaneous inactivation of both enzymatic domains (https://depmap.org/portal/ccle/ accessed on 1 May 2021) [13,14,51,58]. The development of a proteolysis targeting chimera (PROTAC) [59,60,61] degrader of Polθ therefore, represents a plausible therapeutic option since this would fully abolish all of Polθ activities.

## 4. Polθ as a Reverse Transcriptase

Considering that Polθ definitively functions in MMEJ and TLS pathways and suppressing mitotic crossovers and has been implicated in anti-recombination activity and base excision repair, the question arises whether it exhibits additional DNA repair activities that contribute to cancer cell proliferation. A curious characteristic of Polθ-pol is its deficient proofreading function due to acquired mutations (Figure 1A). Pols lacking exonuclease activity typically exhibit low-fidelity DNA synthesis, and in some cases, this enables TLS activity. Interestingly, inactivating the 3′–5′ exonuclease activity of some related A-family bacterial Pol I enzymes allows these polymerases to reverse transcribe RNA like retroviral reverse transcriptases (RTs), which lack proofreading activity [62,63]. Because Polθ is a highly error-prone enzyme that can utilize various double-strand and ssDNA substrates and is void of exonuclease activity such as retroviral RTs, we examined in recent studies whether Polθ-pol can additionally utilize RNA as a template and thus act as an RT [64]. To determine whether Polθ-pol performs RNA-dependent DNA synthesis activity, we chose a biochemical approach. Our studies utilizing radio-labeled DNA/RNA primer templates found that Polθ-pol exhibits robust RT activity, similar to retroviral RTs (Figure 5A) [64]. Notably, Polθ-pol did not require any specific reaction or buffer conditions and performed RNA-dependent DNA synthesis activity on various template constructs and sequences, and its RT activity mimicked retroviral RTs, such as those expressed by Human Immunodeficiency Virus (HIV), Avian Myeoloblastosis Virus (AMV), and Moloney Murine Leukemia Virus (M-MulV). Most remarkably, our recent report found that Polθ-pol exhibits a higher fidelity and velocity of 2′-deoxyribonucleotide incorporation on RNA versus DNA [64]. This suggests the enzyme was selected to be more accurate and efficient on RNA. In contrast to Polθ-pol, all other recombinant human Pols from the A, X, and B families failed to show any RT activity. Y-family Pols κ and η, however, showed minimal RT activity, resulting in the addition of only a few nucleotides on RNA under identical conditions as Polθ-pol and HIV RT, which fully extended DNA/RNA primer-templates.

To probe the mechanism by which Polθ-pol uniquely accommodates DNA/RNA hybrids, we utilized X-ray crystallography to solve the ternary structure of Polθ-pol on a DNA/RNA primer-template with incoming 2′,3′-dideoxyguanosine triphosphate (ddGTP). Remarkably, the 3.2 Å structure of the Polθ-pol:DNA/RNA:ddGTP complex revealed that the thumb domain undergoes a major structural rearrangement to accommodate the DNA/RNA hybrid when compared to the previously solved structures of Polθ-pol binding to a DNA/DNA primer-template (Figure 6A–C) [26]. For example, 57% of residues within this subdomain, which interacts with the hybrid double-helix region five base-pairs upstream from the 3′ primer terminus, undergo conformational changes from helices to loops (Figure 6B). This partial refolding of the thumb domain may be needed to provide the necessary interactions with the DNA/RNA hybrid that is wider than B-form DNA/DNA and takes a slightly different orientation in its interaction with the polymerase (Figure 6B). The 12 Å shift observed for residue K2181 highlights the dramatic reconfiguration of the thumb subdomain (Figure 6D). In contrast to the two previously solved structures of Polθ-pol:DNA/DNA:ddGTP complexes, which were in the closed configuration, our Polθ-pol:DNA/RNA:ddGTP complex was captured in the open configuration. Thus, the O-helix in the fingers subdomain is rotated outward, and the incoming ddGTP substrate is partially solvent-exposed (Figure 6C). The open to closed conformational change upon dNTP binding by A-family Pols on DNA/DNA has been extensively characterized by many laboratories and continues to provide intriguing mechanistic insight into these highly conserved enzymes [65,66,67,68,69]. Another notable shift observed in the Polθ-pol:DNA/RNA:ddGTP complex includes residue E2246 (palm subdomain), which appears to form a specific ribose 2′-hydroxyl interaction with the RNA template (Figure 6E). Additional multiple hydrogen bonds are observed between the 2′-hydroxyl group of the RNA template and Thr/Gln residues of Polθ-pol (Figure 6F). Polθ residue Y2391 also forms a hydrogen bond with the template ribonucleotide that pairs with the incoming ddGTP (Figure 6G). Similar hydrogen bonds are observed in structures of retroviral RTs, indicating a similar binding mechanism on RNA [70,71]. Overall, the major conformational changes observed by Polθ-pol on the DNA/RNA relative to prior structures of the enzyme captured on DNA/DNA are unprecedented and reveal that Polθ-pol possesses extraordinary plasticity, which likely explains its unusual promiscuity relative to other Pols.

Our studies also confirmed the ability of Polθ to utilize ribonucleotide template bases during its DNA repair activities in cells. For instance, we utilized a newly developed MMEJ GFP reporter assay in which MMEJ activity results in activation of a linear GFP expression vector [20]. To probe Polθ RNA-dependent DNA synthesis activity in cells, the GFP reporter was modified to include multiple ribonucleotides adjacent to the microhomology tract (Figure 5B). Here, activation of the GFP expression vector required RNA-dependent DNA repair synthesis activity during MMEJ. Our studies clearly showed that the inactivation of Polθ via CRISPR-Cas9 genetic engineering resulted in a significant reduction in MMEJ, which is consistent with the ability of Polθ to utilize template ribonucleotides during its DNA repair synthesis activity during MMEJ (Figure 5B). Intriguingly, recent studies reveal RNA polymerase II synthesis of RNA at DSBs, which depends on the MRN complex [72,73,74]. Whether Polθ acts on these DSB associated RNA substrates that are involved in the DNA damage response remains unknown.

Being that multiple studies suggest RNA-mediated DNA repair as a potential mechanism of DNA repair in eukaryotic cells [75,76,77], our findings suggest a plausible function for Polθ in tolerating ribonucleotides during DNA repair. In one scenario, unrepaired ribonucleotides at or near DSB ends can serve as template bases during DNA repair synthesis by Polθ during MMEJ (Figure 5B). The possibility also exists that Polθ utilizes ribonucleotides during TLS (Figure 5C). In support of this idea, ribonucleotides are the most frequently occurring lesion in eukaryotic genomes as a result of their misincorporation by replicative Pols [78]. Thus, in the event that genome embedded ribonucleotides are not efficiently repaired, such as in RNASEH2 deficient cells, replication forks are likely to become arrested. This is owing to the fact that replicative Pols are unable to tolerate template ribonucleotides and stall upon encountering these lesions [78,79]. In this scenario, TLS would be a desirable outcome, especially if the ribonucleotide lesions can be replicated accurately. Although future studies will be needed to test Polθ TLS activity opposite genome-embedded ribonucleotides in cells, the enzyme’s ability to easily accommodate A-form DNA/RNA hybrids relative to other Pols reveals its unique and extraordinary structural plasticity. Hence, it will be interesting to determine whether similar structural rearrangements within the enzyme’s thumb subdomain occur during MMEJ in order to accommodate minimally paired ssDNA overhangs in an active configuration within the enzyme’s active site.

## 5. Conclusions and Perspectives

The continued and growing interest in Polθ biology and drug development is expected to lead to many exciting discoveries regarding the respective activities of Polθ helicase and polymerase, as well as inform the development of improved therapeutics targeting this multi-functional enzyme. The recent report on Artios Pharma’s allosteric Polθ-pol inhibitor class demonstrates for the first time that this specific domain of Polθ is indeed druggable. Another recent study indicates that the helicase domain is also druggable. Here, the authors re-purposed the antibiotic Novabiocin as a Polθ-hel inhibitor that exhibited high micromolar IC_50_. Further development of potent and selective Polθ-hel inhibitors will be needed to fully assess the potential of this domain as a drug target in HR-deficient cells. As with all early-stage drugs, it remains to be seen whether Polθ-pol or Polθ-hel inhibitors will lead to clinically effective therapeutics. Notably, decades of drug development research targeting HIV RT and other viral polymerases demonstrate the successful development of selective and potent inhibitors against these polymerases, including both competitive nucleoside prodrugs and allosteric non-nucleoside drug inhibitors [80,81,82,83,84]. For example, the prodrug nucleotide analog remdesivir was recently approved for emergency use against Sars-Cov-2 RNA polymerase [85,86]. Clinical grade drug development against human DNA helicases, however, has yet to be achieved. However, considering that the ATP binding pocket within Polθ-hel is well defined and solvent-exposed like kinases [24], this domain will likely be a viable drug target for clinical-grade candidates specifically designed as Polθ-hel inhibitors.

A major question that arises from the recent research published by Chandramouly et al. is whether Polθ’s RT activity specifically contributes to particular DNA repair functions, such as cellular tolerance to genome embedded ribonucleotides. Separation of function mutations may be needed to pursue such questions. Considering that Polθ’s RT activity appears to be similar to HIV and other retroviral RTs, a question that arises is whether HIV RT nucleoside inhibitors may be useful for targeting Polθ DNA synthesis activity in cells, and thus, can potentially be re-purposed as anti-cancer agents. Indeed, Polθ and other A-family Pols with a conserved tyrosine residue within the fingers subdomain are known to be inhibited by 2′,3′,-dideoxyribonucleoside triphosphates (ddNTPs) [27]. Thus, it would be interesting to determine the effects of nucleoside prodrug inhibitors of HIV RT on Polθ cellular activities and the survival of HR-deficient cells. For instance, prodrugs such as zalcitabine and didanosine are converted by nucleoside/nucleotide kinases into their respective active metabolites, ddCTP and ddATP, which inhibit Polθ. The clinical impact of such nucleoside analogs, however, may be limited due to their modest potency and cross-reactivity against mitochondrial Polγ and possibly other Pols. Nevertheless, based on the successful history of competitive nucleotide inhibitors of viral polymerases, the possibility exists that potent and selective nucleotide analog inhibitors of Polθ-pol can be developed in addition to allosteric inhibitors. Regardless of their mechanism of action, the continued development of Polθ-pol and Polθ-hel drug-like inhibitors is expected to generate a lot of excitement for both the translational and basic research communities.

## Figures and Tables

**Figure 1 genes-12-01146-f001:**
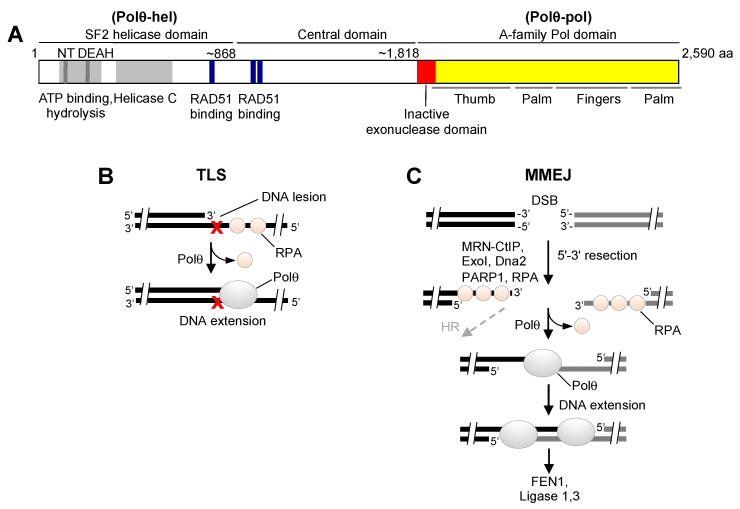
Overview of Polθ structure and function in DNA repair. (**A**) Schematic of full-length Polθ. (**B**) Polθ performs TLS enabling cellular tolerance to DNA damaging agents, such as ultraviolet light. (**C**) Polθ promotes MMEJ, which results in the error-prone repair of DSBs and resistance to ionizing radiation and topoisomerase inhibitors.

**Figure 2 genes-12-01146-f002:**
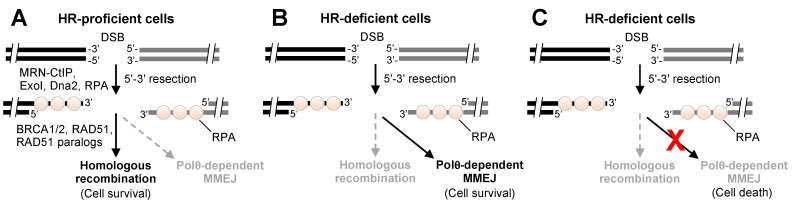
Model of Polθ dependency in HR-deficient cells. (**A**) DSBs are predominantly repaired in S/G2 cell cycle phases by accurate HR, which is promoted by BRCA1/2, RAD51, and associated proteins. Polθ-dependent MMEJ serves as a backup DSB repair pathway that acts on 3′ ssDNA overhangs and causes large indels. (**B**) HR-deficient cells rely on Polθ-dependent MMEJ for DSB repair and cell survival. (**C**) Inactivation or suppression of Polθ causes synthetic lethality in HR-deficient cells as a result of insufficient DNA repair.

**Figure 3 genes-12-01146-f003:**
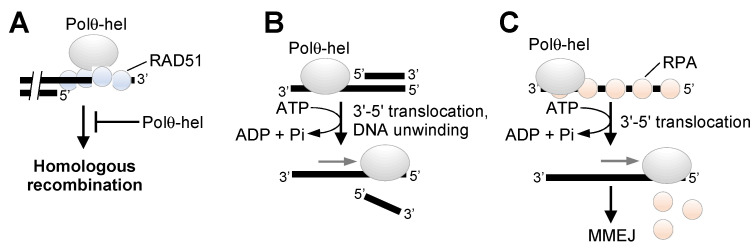
Biochemical mechanisms of Polθ helicase. (**A**) Polθ-hel was reported to interact with RAD51 and suppress HR and toxic RAD51:DNA intermediates via its ATPase activity. (**B**) Polθ-hel was shown to unwind short double-strand DNA with 3′–5′ polarity in an ATP hydrolysis-dependent manner. (**C**) Polθ-hel was shown to dissociate RPA from ssDNA in an ATP hydrolysis-dependent manner, and evidence indicates that this activity stimulates MMEJ.

**Figure 4 genes-12-01146-f004:**
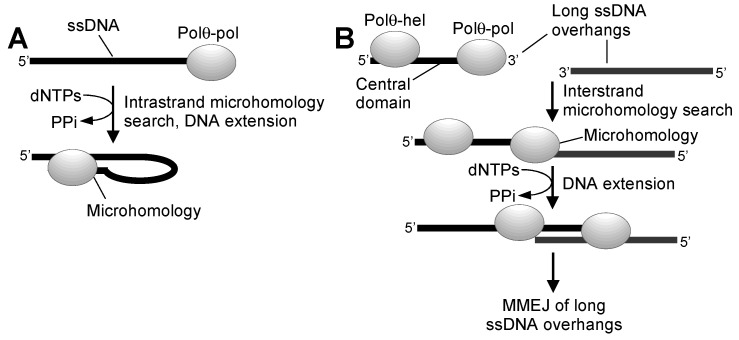
Full-length Polθ promotes MMEJ of long 3′ ssDNA overhangs. (**A**) Polθ-pol promotes snap-back replication on ssDNA by intramolecular microhomology search, transient microhomology annealing, and subsequent extension of the minimally paired 3′ terminal end of ssDNA. This activity suppresses MMEJ of long ssDNA overhangs by Polθ-pol. (**B**) Full-length Polθ performs MMEJ of long ssDNA. Covalent attachment of Polθ-hel to Polθ-pol through the Polθ central domain or a short peptide linker suppresses intramolecular microhomology search and snap-back replication, which results in promoting intermolecular microhomology search and subsequent MMEJ.

**Figure 5 genes-12-01146-f005:**
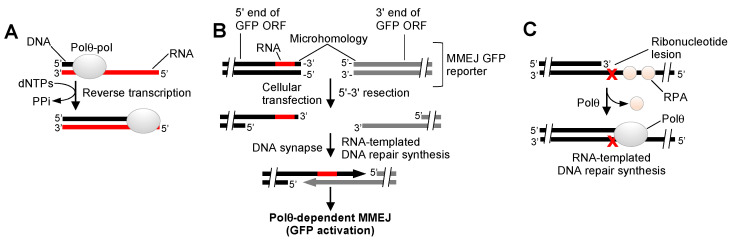
Polθ-pol RNA-dependent DNA synthesis activities. (**A**) Polθ-pol reverse transcribes RNA, similar to retroviral reverse transcriptases. (**B**) Schematic of an MMEJ GFP DNA reporter construct used to detect Polθ RNA-templated DNA repair synthesis in cells. (**C**) Model of Polθ TLS opposite a ribonucleotide lesion.

**Figure 6 genes-12-01146-f006:**
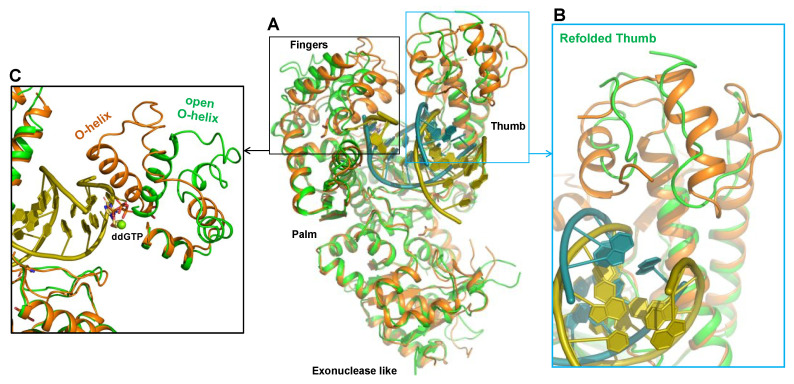
Structure of the Polθ-pol:DNA/RNA:ddGTP ternary complex. (**A**) Overlap of the structures of Polθ-pol:DNA/RNA:ddGTP (green) and Polθ-pol:DNA/DNA:ddGTP (orange, PDB: 4 × 0 q). The fingers and thumb domains (boxed in black and blue) differ in the two structures. (**B**) The thumb domain of Polθ-pol in the DNA/DNA has typical α-helix folding (orange), but the helices of the thumb of Polθ-pol in complex with DNA/RNA is partially re-folded to loops (green). (**C**) The O-helix of the fingers of Polθ-pol in complex with DNA/DNA has a closed state (orange), but the fingers of Polθ-pol in complex with DNA/RNA have an open state (in green). (**D**) The conformational shift of K2181 of the thumb domain between the closed Polθ-pol:DNA/DNA and the open Polθ-pol:DNA/RNA structures. (**E**) In the Polθ-pol:DNA/RNA structure, E2246 forms a hydrogen bond with the 2′-OH of the RNA template. (**F**) Additional hydrogen bond interactions between 2′-OH groups of the RNA template and Polθ-pol residues. Zoomed-in images of these bonding interactions are shown on the left. (**G**) The main-chain carbonyl of Y2931 in Polθ-pol forms a hydrogen bond with the 2′-OH of the template ribonucleotide that pairs with the incoming ddGTP site in the active.

## Data Availability

Not applicable.

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
