# Peer review of "DNA Polymerase θ: A Cancer Drug Target with Reverse Transcriptase Activity"

_genes, 2021, doi:10.3390/genes12081146_

Round 1

Reviewer 1 Report

This Review article on the DNA Polymerase Theta is a comprehensive description of the known activities of Theta focusing on MMEJ and recent novel results from the authors lab showing a reverse transcriptase, or NTP incorporation, activity for Theta. The figures are clear and important as the reader moves through the text. In all, this is a valuable contribution to the field. I only have a few minor comments.

1) End of section 2. Maybe a little more explanation on why Theta is SL with HJ resolvases. Is it that the hypothesis that the HR intermediates that Theta acts on are then converted to MMEJ products?

2) Figure 6 looks as though D,E,F,G are on top of the figure. Is this as intended? It is a little unorthodox and difficult to follow. Would suggest moving A,B,C to the top.

3) Conclusions. I am wonder what drug screens have been tried. For example, have NRTIs been tried for Theta or is this just a suggestions. Also ProTacs as mentioned on Page 5 (line 195). This sentence is written as they have been tried on Theta, but I believe these references are just for the general Protac strategy. Maybe clarify or reword?

4) This comment is a little unfair, but in light of a very recent publication Zatreanu et al.  PMID: 34140467, I think the authors should take some space to address these new inhibitors ART558 for Theta. They appear to disrupt MMEJ and are SL with B1 or B2 mutant cells, while enhancing PARPi sensitivity. This is really interesting with regards to this review and the authors conclusions. I think readers would appreciate the authors perspective here, and the implications of future drugs based on this initial design.

Reviewer 2 Report

The review is giving an overview over polymerase θ any why it may be used as a drug target. (on the front webpage of mdpi it is named polymerase Q, what might be a typo/problem with Greek alphabet...in the pdf it is θ ...or is this a  because of POLQ as gene for the protein polθ, then it should be made consistent...in the whole text is a mixture of Polθ, Polq, POLQ)

line 83 correct for space

line 89 "CtIP" please give definition here

line 109 Pol-pol and Pol-hel are not introduced in the text previously, they appear in Fig. 1 A and may discussed/introduced right there, ...overall may be some words for explanation what is important (may reappear) in Fig. 1 A

line 116 "SL" please explain

line 218 any data showing higher fidelity and velocity?

regarding drug approach

maybe citing of

Tomasso et al.

Curr Mol Med. 2014 Jan;14(1):96-114.
doi: 10.2174/15665240113136660080.
What makes y family pols potential candidates for molecular targeted therapies and novel biotechnological applications 
